# Procalcitonin Predicts Bacterial Infection, but Not Long-Term Occurrence of Adverse Events in Patients with Acute Coronary Syndrome

**DOI:** 10.3390/jcm11030554

**Published:** 2022-01-22

**Authors:** Rita Pavasini, Gioele Fabbri, Federico Marchini, Nicola Bianchi, Maria Angela Deserio, Federico Sanguettoli, Filippo Maria Verardi, Daniela Segala, Graziella Pompei, Elisabetta Tonet, Matteo Serenelli, Serena Caglioni, Gabriele Guardigli, Gianluca Campo, Rosario Cultrera

**Affiliations:** 1UO Cardiologia, Azienda Ospedaliero Universitaria di Ferrara, 44124 Ferrara, Italy; gioele.fabbri2@gmail.com (G.F.); federico.marchini@edu.unife.it (F.M.); nbianchi95@gmail.com (N.B.); maria_angela.93@hotmail.it (M.A.D.); f.sanguettoli@gmail.com (F.S.); filippomaria.verardi@gmail.com (F.M.V.); graziella.pompei@outlook.it (G.P.); tonet.elisabetta@gmail.com (E.T.); matteoserenelli@gmail.com (M.S.); s.caglioni@studenti.uniba.it (S.C.); gabri.guardigli@libero.it (G.G.); cmpglc@unife.it (G.C.); 2UO Malattie Infettive, Azienda Ospedaliero Universitaria di Ferrara, 44124 Ferrara, Italy; daniela.segala@unife.it (D.S.); rosario.cultrera@unife.it (R.C.)

**Keywords:** procalcitonin, infection, acute coronary syndrome, mortality, myocardial infarction

## Abstract

This study compiles data to determine if procalcitonin (PCT) values may predict both the risk of bacterial infection and potentially negative long-term outcomes in patients with acute coronary syndromes (ACS). All patients with a diagnosis of ACS that had PCT levels assessed during the first 24 h of hospitalization were enrolled in this study. The primary outcome was to detect the presence of bacterial infection defined as the occurrence of fever and at least one positive blood or urinary culture with clinical signs of infection. The secondary outcome was to monitor the occurrence after 1 year of the composite outcome of all-cause mortality, stroke and myocardial infarction. Overall, 569 patients were enrolled (mean age 69.37 ± 14 years, 30% females). Of these, 44 (8%) met the criteria for bacterial infection. After multivariate analysis, PCT and SBP were found to be independent predictors of bacterial infections (OR for PCT above the cut-off 2.67, 95% CI 1.09–6.53, *p* = 0.032 and OR for SBP 0.98, 95% CI 0.97–0.99, *p* = 0.043). After 1 year, the composite outcome of all-cause death, MI and stroke occurred in 104 patients (18%). PCT was not found to be an independent predictor of these outcomes. In conclusion, when assessing ACS, we found that testing for PCT levels during hospital admissions procedures was a good predictor of bacterial infections but not of all-cause mortality, stroke, or myocardial infarction. Clinicaltrial.org identifier: NCT02438085.

## 1. Introduction

Procalcitonin (PCT) is a precursor of the hormone calcitonin produced by thyroid c-cells. In healthy people, PCT has a very low plasma level. It follows that PCT is a known marker of bacterial infection because after endotoxin stimulation, plasma concentrations in PCT rise, due to the increase in its production, which is facilitated by hepatocytes and immune cells [1]. Usually, the PCT levels rise in relation to the severity of an infection and remain elevated for the duration of the infection [2]. However, PCT is also thought to be an acute phase protein, which possibly causes plasma levels to rise in cases of sterile inflammation. For this reason, its role would be twofold: as a marker of infection (overt or not); and as a prognosticator in cases of generic inflammation [2] (e.g., acute pancreatitis, chronic obstructive pulmonary disease, or stroke). Therefore, PCT has also been investigated to potentially diagnose inflammation as it relates to acute coronary syndromes (ACS), but its role remains controversial [3,4,5,6]. It has been shown that PCT acutely increases in patients with myocardial infarction (MI), but mainly in patients with concomitant complications such as cardiogenic shock or pulmonary edema [7,8,9,10]. Indeed, patients with MI often present with fever and dyspnea, fatigue, and/or increased white blood cell counts in the acute phase. These are all signs and symptoms usually related to heart failure complicating ACS, but if misinterpreted they might induce an overuse of antibiotics in the absence of a real bacterial infection [1,4]. As a matter of fact, PCT levels increase only in a small percentage of uncomplicated ST elevation myocardial infarction (STEMI) occurrences; and in very few cases of non-ST elevation myocardial infarction (NSTEMI) and unstable angina [1]. Whether an early assessment of PCT levels in ACS is also related to bacterial infection has been less investigated, nor is it fully understood how this biomarker is related to adverse outcomes in ACS patients. Therefore, the aims of the present analysis are to verify in a large population of ACS patients if procalcitonin remains an independent predictor of infection during an acute cardiac event and can predict the likelihood of worsening outcomes after one year.

## 2. Materials and Methods

The Acute coRonary sYndrOmes proSpective regisTry Of Ferrara (ARYOSTO) is a prospective, single-center study collecting data about baseline characteristics, treatment, and outcomes of all patients admitted to the University Hospital of Ferrara with a diagnosis of ACS. The ARYOSTO study started on May 2015 and is still ongoing. For the present purpose, we considered patients admitted to the hospital from June 2018 to December 2019. The study is registered on clinicaltrials.gov with the identifier NCT02438085. The study was conducted in accordance with the ethical principles of the Declaration of Helsinki. Patients were informed that their participation was voluntary and all gave written consent.

### 2.1. Study Population

Inclusion criteria were diagnosis of ACS (ST-elevation MI, non-ST-elevation MI, and unstable angina) according to current guidelines [11,12] and concentration of procalcitonin in the blood sample collected upon admission at the emergency room or cardiology ward (institutional protocol started on June 2018). Of note, only Type 1 MI patients were included in this analysis according to the fourth definition of MI [13]. Exclusion criteria were cardiogenic shock at presentation, known active or chronic infections prior to index hospitalization, chronic inflammatory diseases, severe hepatic or renal dysfunction, malignancies, inability to guarantee follow-up and refusal to provide consent. The management of participants was at the discretion of attending physicians and followed institutional protocols and international guidelines [11,12].

### 2.2. Procalcitonin

On admission, peripheral venous blood was drawn from the antecubital vein and four routine blood tests were performed including blood count, renal and hepatic function assessment, high sensitivity troponin I, brain natriuretic peptide, and PCT tests. Plasma level of PCT was assessed using Roche Cobas clinical and immunochemistry test kits (Roche, Swiss) and a Roche MODULAR E170 automatic electrochemiluminescence immunoassay analyzer (Roche, Swiss). The upper cut-off for normal levels of PCT was 0.5 ng/mL.

### 2.3. Data Collection

All clinical, treatment and outcome data were prospectively collected using a dedicated electronic case report form (eCRF). Specialized personnel performed this procedure. The eCRFs were periodically monitored and verified with source data. The following data were prospectively collected: anthropometric data, cardiovascular (CV) risk factors, CV history and comorbidities, laboratory data, CV drugs administered during hospitalization, procedural details including descriptions of the extension of coronary artery disease, interventional strategies, stent placements, procedural and management complications, and access site as well as in-hospital adverse events.

### 2.4. Study Endpoints

The primary outcome of the study was to identify the occurrence of bacterial infection defined by the presence of fever and at least one positive blood or urinary culture with clinical signs of infection (e.g., positive chest X-ray, positive urinary exam, clinical evaluation by an infectious disease specialist and routine antibiotic prescription) [14]. The secondary outcome was the composite endpoint of considering all-cause mortality, stroke and myocardial infarction events after one year of follow-up. The diagnosis of myocardial infarction required the detection of a combination of symptoms, including electrocardiographic changes and significant increases in cardiac markers (e.g., troponin). Cerebrovascular accidents were defined as the clinical diagnosis of stroke and transient ischemic attacks. Adverse events were adjudicated by a clinical events committee that reviewed the original source documents.

### 2.5. Patient and Public Involvement Statement

Patients and members of the public were not involved in the design, conduct, reporting, or dissemination plans of this study.

### 2.6. Statistical Analysis

Previous data about the incidence of bacterial infection during ACS are controversial with rates ranging between 4% and 15% [15,16]. We set an occurrence rate of bacterial infection around 6% and supposed that the occurrence of bacterial infection would be at least 3 times higher in ACS patients with baseline (before the clinical signs of infection) procalcitonin above the cut-off for normal levels as compared to those with procalcitonin in the range of normality. Then, we determined that at least 550 patients with complete data were needed. Continuous data were tested for normal distribution with the Kolmogorov–Smirnov test. Normally distributed values were presented as mean ± SD and compared to a t-test. Otherwise, median, interquartile range, and Mann–Whitney U tests were applied. Categorical variables were summarized in terms of counts and percentages and were compared by using the two-sided Fisher’s exact test. Variables analyzed were stratified according to PCT levels above or below the cut-off considered normal. Univariate and multivariate logistic regression analysis was performed to test predictors of bacterial infection. Only variables included in Table 1 with *p* < 0.1 after univariate analysis were included in multivariate analysis. The area under the receiver operating characteristic (ROC) curve (AUC) and Youden index were analyzed to find the best cut-off value of PCT to predict the primary outcome. Univariate and multivariate Cox regression analysis was also performed to identify predictors of the composite outcome. Two models were performed: in one PCT was considered together with other variables found statistically significant at univariate analysis, and in the second model the occurrence of infection was used instead of PCT. All statistical analyses were performed with Stata/SE version 16 software (Stata Corp, College Station, TX, USA).

## 3. Results

From June 2018 to December 2019, a total of 1004 patients with a diagnosis of ACS were admitted. Overall, 569 (57%) patients fulfilled our inclusion and exclusion criteria and were considered for the present analysis (Table 1).

Three hundred nine patients (54%) were admitted for ST-segment elevation MI. Overall, 565 (99%) underwent invasive surgery and received coronary artery angiography, whereas 458 (80%) were treated with percutaneous coronary revascularization, and no one received surgical coronary revascularization (e.g., coronary artery bypass graft).

The median level of PCT was 0.05 (0.02–0.13) ng/mL. Overall, 506 (89%) patients showed a value in the normal range, whereas 63 (11%) did not. Population characteristics stratified according PCT level above and below the cut-off of normality are summarized in Table 1.

Patients with higher PCT were significatively older (77 ± 12 vs. 68 ± 14 years, *p* < 0.001) and diabetics (46% vs. 25%, *p* < 0.001) although they were less frequently smokers or former smokers (42% vs. 57%, *p* = 0.023). In these patients, heart rate (HR) at baseline was higher (102 ± 26 vs. 82 ± 22 bpm, *p* < 0.001) and systolic blood pressure (SBP) was lower (129 ± 29 vs. 139 ± 28 mmHg, *p* = 0.007).

In patients with PCT above the cut-off, STEMI was less common as a clinical presentation (37% vs. 57%, *p* = 0.003) and invasive management strategies were less applied (92% vs. 99%, *p* < 0.001). Hemoglobin was lower (12.2 ± 2.5 vs. 13.5 ± 2.1 g/dL, *p* < 0.001), platelets higher (269 ± 114 vs. 235 ± 74 × 10^3^/mmc, *p* = 0.002) with a significatively lower estimated glomerular filtration rate (49 ± 27 vs. 76 ± 25 mL/min, *p* < 0.001).

### 3.1. Primary Outcome

Overall, 44 (8%) patients met the criteria for bacterial infection. Of these, 44 (100%) have positive blood culture and 20 (45%) positive urinary culture. Gram positive germs were the most frequently isolated. Procalcitonin values in patients developing bacterial infections were higher compared to others (0.16 vs. 0.05 ng/mL, *p* < 0.001), with 36 (81%) patients having PCT above the cut-off level. Age; gender; smoking habits; HR and SBP; and rates of heart failure after admission, coronary angiography, hemoglobin, estimated glomerular filtration (eGFR) and PCT above the cut-off value, were predictors of the outcome (Table 2).

After multivariate analysis, only PCT above the cut-off and SBP resulted as independent predictors of bacterial infections (OR for PCT above the cut-off 2.67, 95% CI 1.09–6.53, *p* = 0.032, and OR for SBP 0.98, 95% CI 0.97–0.99, *p* = 0.043). The area under the ROC curve of PCT for infection was 0.68 (95% CI 0.63–0.71) (Figure 1), with a best cut-off of 0.165 (sensitivity 50%, and specificity 82%).

### 3.2. Secondary Outcome

After 1 year, the composite outcome of all-cause death, MI and stroke occurred in 104 patients (18%). The secondary outcome occurred significantly more in patients with PCT levels above the cut-off (28% vs. 7%, *p* < 0.001). Table 3 describes variables significantly related to the secondary outcome by univariate analysis. After multivariate analysis, PCT above cut-off did not result as an independent predictor of the outcome (model 1 in Table 3, HR 1.31, 95%CI 0.73–2.34, *p* = 0.99). To further support the findings, we repeated the analysis of bacterial infections despite PCT levels above the cut-off (Model 2 in Table 3). We found that bacterial infection did not emerge as an independent predictor of 1-year outcomes (HR = 0.94, 95% CI 0.48–1.93, *p* = 0.85).

## 4. Discussion

The main finding of this study was that PCT concentrations above the cut-off levels remain a predictor of bacterial infection also in patients hospitalized with ACS. At the same time, this study confirmed that even if PCT levels tend to be higher in patients who experience cardiogenic shock during hospitalization, this is not a true marker of worse cardiovascular outcomes at 1-year follow-up.

Reindl et al. already showed in a cohort of 141 STEMI patients undergoing percutaneous coronary intervention, that there was not an association between PCT levels at 24 and 48 h after intervention and myocardial and microvascular injury as detected by cardiac magnetic resonance [7]. Authors concluded that PCT might be a necrosis-independent clinical marker that might be useful to detect infection during ACS [7]. Our data confirms these finding, showing the usefulness of performing early PCT diagnostics on ACS patients to detect the development of bacterial infections in patients still without signs of infection. A previous study on 230 MI patients showed that PCT was useful for ruling out infection in patients hospitalized with MI [15]. The authors suggested that PCT could serve as a valid auxiliary test useful for ruling out infections in patients with MI with a negative predictive value of nearly 99% [15]. Our study expands on previous findings by focusing on a larger population (*n* = 569) and including all types of ACS. The rate of infection we found (eight percent) was lower than that of Vitkon-Barkay et al. [15]. This might be related to the slight difference in the definition used to detect infection [15]. In our study, the inclusion of patients with a fever and positive cultural tests together, combined with clinical and laboratory data and the infectious specialist evaluation, might have made the inclusion of patients with an infectious event more restrictive.

The Canada Acute Coronary Syndrome (C-ACS) score has been proposed to identify patients at risk of infection after MI. Variables considered in the calculation of this score were age ≥ 75 years, Killip class > 1, systolic blood pressure < 100 mmHg and heart rate > 100 beats/min [17]. In our study, patients with higher PCT level were older, presented more often with acute heart failure, had lower blood pressure, and higher HR. However, in multivariate analyses that also considered PCT, only the last variable and SBP were seen to be independent predictors of infections in ACS.

Previous studies have shown that in patients with ACS complicated by cardiogenic shock, PCT is usually elevated and this more often predicted in-hospital mortality, even if with a low correlation strength [18]. The mechanism related to the increase in PCT blood levels in cardiogenic shock is related to the exposure to bacterial endotoxins due to bowel ischemia and alteration of gut permeability. In these critically ill patients, a dynamic assessment of PCT was more useful than a static approach [1] considering that it was shown that static PCT measurements at 24 h after the onset of ST-elevation MI complicated by cardiogenic shock were not an independent predictor of in-hospital mortality [19]. By contrast, our data shows that one systematic measurement of PCT, usually at the first day of admission after an ACS event, is predictive of the development of infection, while it is not related to the long-term outcome of patients. Senturk et al. already showed that in patients with ACS that are not in cardiogenic shock, even if level of PCT might be increased, it is not related to coronary artery disease severity nor to 3-month mortality [9]. Therefore, our data confirms previous findings about the role of PCT in predicting infections even in the setting of clinical presentations of ACS, without being a prognosticator of adverse outcomes.

### Study Limitations

The main limitation of this study was related to the absence of data identifying the dosage of PCT at different time points. Secondly, the population analyzed, while larger than that of the previous study we consulted on the same issue [2,3,4,5,6,7,8,9], was still limited to 569 ACS patients. Moreover, there was a lack of correlation between PCT and imaging data (e.g., echocardiographic findings or cardiac magnetic resonance data). Finally, the low number of cardiovascular events in the follow-up period did not allow for an adequate evaluation of the impact of PCT on the single components of the secondary composite outcome.

## 5. Conclusions

In conclusion, in a large cohort of patients with ACS, the systematic assessment of PCT during hospital admission is related to the development of bacterial infection but it is not associated with the long-term composite outcome of all-cause mortality, stroke, and myocardial infarction.

## Figures and Tables

**Figure 1 jcm-11-00554-f001:**
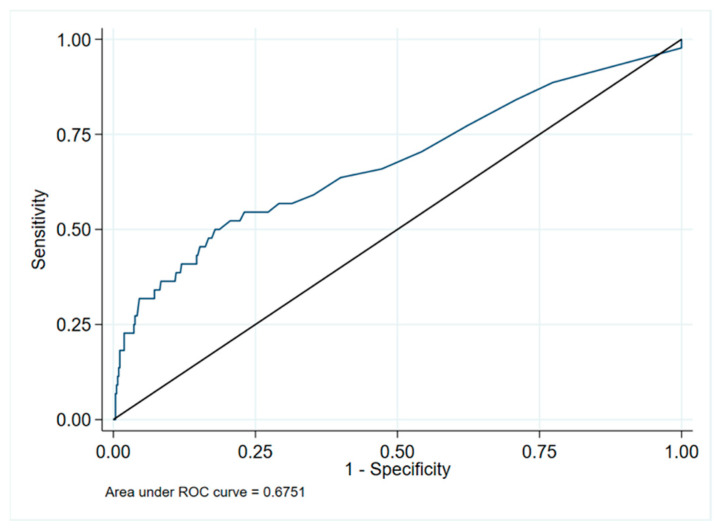
ROC curve for infection and PCT. ROC—receiver operating characteristics; PCT—procalcitonin.

**Table 1 jcm-11-00554-t001:** Population characteristics (overall and stratified), for having a procalcitonin at a baseline above the cut-off value.

	Total *n* = 569	PCT below Cut-Off *n* = 506	PCT above Cut-Off *n* = 63	*p* Value	No Infection *n* = 525	Infection *n* = 44	*p* Value
Age at baseline, mean ± sd	69.37 ± 13.80	68.47 ± 13.71	76.59 ± 12.36	<0.001	68.85 ± 13.70	75.56 ± 13.57	0.002
BMI—Kg/mq, median (IQR)	26.12 (23.78–29.35)	26.26 (24.02–29.38)	24.91 (22.56–27.4)	0.009	26.12 (23.96–29.06)	25.93 (22.89–30.47)	0.55
Female sex, *n* (%)	170 (30)	145 (29)	25 (40)	0.071	150 (29)	20 (45)	0.019
History							
Smoking habit, *n* (%)	310 (55)	285 (57)	25 (42)	0.023	297 (57)	13 (31)	<0.001
Hypertension, *n* (%)	377 (67)	330 (65)	47 (76)	0.10	344 (66)	33 (77)	0.14
Dyslipidemia, *n* (%)	270 (49)	240 (49)	30 (51)	0.76	250 (49)	20 (49)	0.98
Diabetes mellitus, *n* (%)	155 (27)	126 (25)	29 (46)	<0.001	140 (27)	15 (34)	0.29
CKD, *n* (%)	70 (51)	47 (47)	23 (62)	0.11	56 (49)	14 (61)	0.30
Baseline characteristics							
Heart rate, mean ± sd	83.95 ± 23.13	81.78 ± 21.85	101.61 ± 25.76	<0.001	83.29 ± 22.50	91.73 ± 28.78	0.020
Systolic blood pressure, mean ± sd	138.25 ± 28.30	139.36 ± 28.09	129.07 ± 28.60	0.007	139.28 ± 28.03	126.09 ± 28.91	0.003
Cath lab							
ST-elevation, *n* (%)	309 (54)	286 (57)	23 (37)	0.003	289 (55)	20 (45)	0.22
Coronary angiography, *n* (%)	564 (99)	505 (100)	59 (94)	<0.001	523 (99)	41 (93)	<0.001
Percutaneous coronary intervention	458 (80)	413 (82)	45 (71)	0.054	427 (81)	31 (70)	0.08
Heart failure after admission, *n* (%)	46 (8)	30 (6)	16 (25)	<0.001	35 (7)	11 (25)	<0.001
Laboratory							
White blood cells—×10^3^/mmc, median (IQR)	10.32 (8.00–12.93)	10.04 (7.88–12.37)	13.80 (10.77–16.74)	<0.001	10.29 (7.89–12.71)	10.92 (8.61–15.98)	0.058
Hemoglobin—g/dL, mean ± sd	13.31 ± 2.14	13.45 ± 2.06	12.22 ± 2.47	<0.001	13.42 ± 2.09	12.03 ± 2.37	<0.001
Platelets—×10^3^/mmc, mean ± sd	238.70 ± 80.27	234.98 ± 74.39	268.73 ± 113.95	0.002	238.71 ± 79.80	238.56 ± 86.86	0.99
eGFR—mL/min, mean ± sd	72.69 ± 26.56	75.64 ± 25.00	48.78 ± 26.94	<0.001	73.93 ± 26.29	56.94 ± 25.17	<0.001
Hs troponin peak ng/L, median (IQR)	6000.00 (1061.00–25,868.00)	6712.00 (1061.00–27,398.0)	3806.50 (443.00–22,017.00)	0.17	6191.50 (1065.00–26,325.00)	4593.00 (220.00–19,575.00)	0.23
PCT at baseline above cutoff, *n* (%)	63 (11)	--	--	--	47 (9)	16 (36)	<0.001
PCT ng/mL, median (IQR)	--	--	--	--	0.05 (0.02–0.12)	0.16 (0.04–1.40)	<0.001

PCT—procalcitonin; BMI—body mass index; CAD—coronary artery disease; AF—atrial fibrillation; CKD—chronic kidney disease; PCI—percutaneous coronary intervention; AV—atrio-ventricular; eGFR—estimated glomerular filtration rate; Hs—high sensitivity.

**Table 2 jcm-11-00554-t002:** Uni- and multivariate analysis for bacterial infection.

	Univariate			Multivariate		
	OR	95% CI	*p* Value	OR	95% CI	*p* Value
Age at baseline	1.04	1.01–1.07	**0.002**	1.01	0.98–1.04	0.558
Female sex	2.08	1.12–3.88	**0.021**	1.05	0.49–2.27	0.897
Smoking habit	0.33	0.17–0.65	**0.001**	0.55	0.26–1.20	0.133
Baseline HR	1.01	1.00–1.03	**0.022**	1.00	0.99–1.02	0.520
Baseline SBP	0.98	0.97–0.99	**0.003**	0.99	0.97–1.00	**0.043**
Coronary angiography	0.24	0.06–0.91	**0.037**	0.84	0.13–5.44	0.854
Heart failure after admission	4.67	2.17–10.02	**<0.001**	2.39	0.93–6.17	0.071
White blood cells	1.00	1.00–1.00	0.836			
Hemoglobin	0.76	0.67–0.87	**<0.001**	0.87	0.73–1.02	0.092
GFR	0.98	0.97–0.99	**<0.001**	0.99	0.98–1.01	0.526
PCT above cutoff	5.81	2.93–11.51	**<0.001**	2.67	1.09–6.53	**0.032**

HR—heart rate; SBP—systolic blood pressure; GFR—glomerular filtration rate; PCT—procalcitonin.

**Table 3 jcm-11-00554-t003:** Univariate and multivariate analyses for the secondary composite outcome of all-cause death, stroke and myocardial infarction.

	Univariate			Multivariate Model 1			Multivariate Model 2		
	HR	95% CI	*p* Value	HR	95% CI	*p* Value	HR	95% CI	*p* Value
**Age at baseline**	1.08	1.06–1.10	**<0.001**	1.07	1.04–1.10	**<0.001**	**1.07**	**1.04–1.09**	**<0.001**
**BMI**	0.95	0.91–1.00	**0.048**	0.97	0.92–1.03	0.375	0.97	0.092–1.02	0.252
**Female sex**	1.82	1.22–2.72	**0.003**	0.79	0.47–1.33	0.374	0.79	0.48–1.33	0.378
**Smoking habit**	0.62	0.41–0.94	**0.023**	0.90	0.55–1.45	0.655	0.88	0.54–1.42	0.597
**Hypertension**	1.80	1.12–2.89	**0.016**	0.74	0.43–1.28	0.285	0.74	0.43–1.29	0.291
**Diabetes mellitus**	2.11	1.42–3.14	**<0.001**	1.29	0.76–2.18	0.347	1.30	0.77–2.20	0.320
**Baseline HR**	1.01	1.00–1.02	**0.046**	1	0.99–1.01	0.636	1	0.99–1.01	0.778
**Baseline SBP**	0.99	0.98–1.00	**0.004**	0.99	0.99–1.00	0.127	0.99	0.99–1	0.124
**Coronary angiography**	0.21	0.10–0.45	**<0.001**	0.39	0.13–1.16	0.091	0.40	0.13–1.18	0.095
**Heart failure after admission**	4.90	3.09–7.77	**<0.001**	3.27	1.82–5.88	**<0.001**	0.354	1.96–6.38	**<0.001**
**Bacterial infection**	2.85	1.69–4.81	**<0.001**	--	--	--	0.94	0.48–1.93	0.846
**Hemoglobin**	0.79	0.72–0.86	**<0.001**	1.03	0.93–1.16	0.541	1.03	0.92–1.15	0.584
**Platelets**	1.00	1.00–1.01	**0.003**	1.00	1.00–1.01	**<0.001**	1	1.00–1.01	**<0.001**
**eGFR**	0.97	0.97–0.98	**<0.001**	0.99	0.98–1.00	**0.039**	0.99	0.98–1	**0.019**
**PCT above cutoff**	3.91	2.54–6.04	**<0.001**	1.31	0.73–2.34	0.361	--	--	--

Model 1: PCT above the cut-off was used. Model 2: infection was used instead of PCT. BMI—body mass index; HR—heart rate; SBP—systolic blood pressure; GFR—glomerular filtration rate; PCT—procalcitonin.

## Data Availability

The data that support the findings of this study are available from the corresponding author, (RP), upon reasonable request.

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
