# Peer review of "Procalcitonin Predicts Bacterial Infection, but Not Long-Term Occurrence of Adverse Events in Patients with Acute Coronary Syndrome"

_jcm, 2022, doi:10.3390/jcm11030554_

Round 1
Reviewer 1 Report
In general,
The manuscript entitled " Procalcitonin predicts bacterial infection, but not long-term occurrence of adverse events in patients with acute coronary syndrome " is a interesting characterization of the use to verify in a large population of acute coronary syndrome if procalcitonin remains an independent predictor of infection during an acute cardiac event and if it is related to worse outcome in the follow-up.
The Instrument is well described, the Materials and Methods are clear and the Conclusions are valuable and useful for many in the field interested in the analysis.
"Therefore, PCT has been investigated also in the setting of inflammation related to acute coronary syndromes (ACS) but remaining controversial. It has been showed that PCT acutely increases in patients with myocardial infarction (MI), but mainly in patients with concomitant complications as cardiogenic shock or pulmonary edema. As a matter of fact, only PCT levels increase only in a small percentage of uncomplicated ST elevation myocardial infarction (STEMI) and in very few non-ST elevation myocardial infarction (NSTEMI) and unstable angina. Whether an early assessment of PCT levels in ACS is also related to bacterial infection has being less investigated as well as if this biomarker is related to adverse outcome of ACS patients. Therefore, the aim of the present analysis is to verify in a large population of ACS if procalcitonin remains an independent predictor of infection during an acute cardiac event and if it is related to worse outcome in the follow-up. "
I believe and think that it could be valuable to insert by rephrasing as follows:
Therefore, PCT has been investigated also in the setting of inflammation related to acute coronary syndromes (ACS) but remaining controversial .... [here briefly ... mention why the controversy, for example, how does the ACS (signs and symptoms, mentioning associated risk factors, among others), understanding that ACS should be mentioned because at the end of this paragraph emphasis is placed on them, in the same sense, mention in which cases the PCT is used, also taking up everything.
As mentioned in more detail in the discussion, this may help to better understand this important clinical approach] ... the aim of the present analysis is to verify in a large population of ACS if procalcitonin remains an independent predictor of infection during an acute cardiac event and if it is related to worse outcome in the follow-up.
Even at the beginning of the Introduction, it should be mentioned in general how common the presence of SCD are, this would give a justification that could round off the importance of studying and reporting these cardiovascular cases in this approach related with procalcitonin.
Author Response
Response to Reviewers
Manuscript: jcm-1524997
Title: Procalcitonin predicts bacterial infection, but not long-term occurrence of adverse events in patients with acute coronary syndrome
Corresponding Author: Rita Pavasini, pvsrti@unife.it
Current revision: #1
Reviewer #1
Comment #1
In general,
The manuscript entitled " Procalcitonin predicts bacterial infection, but not long-term occurrence of adverse events in patients with acute coronary syndrome " is a interesting characterization of the use to verify in a large population of acute coronary syndrome if procalcitonin remains an independent predictor of infection during an acute cardiac event and if it is related to worse outcome in the follow-up.
The Instrument is well described, the Materials and Methods are clear and the Conclusions are valuable and useful for many in the field interested in the analysis.
Reply # 1
We really thank the Reviewer #1 for taking the time to review our submission and for the positive comments. We appreciated the comments raised and we prepared an amended version of the manuscript following your suggestions.
Comment #2
"Therefore, PCT has been investigated also in the setting of inflammation related to acute coronary syndromes (ACS) but remaining controversial. It has been showed that PCT acutely increases in patients with myocardial infarction (MI), but mainly in patients with concomitant complications as cardiogenic shock or pulmonary edema. As a matter of fact, only PCT levels increase only in a small percentage of uncomplicated ST elevation myocardial infarction (STEMI) and in very few non-ST elevation myocardial infarction (NSTEMI) and unstable angina. Whether an early assessment of PCT levels in ACS is also related to bacterial infection has being less investigated as well as if this biomarker is related to adverse outcome of ACS patients. Therefore, the aim of the present analysis is to verify in a large population of ACS if procalcitonin remains an independent predictor of infection during an acute cardiac event and if it is related to worse outcome in the follow-up. "
I believe and think that it could be valuable to insert by rephrasing as follows:
Therefore, PCT has been investigated also in the setting of inflammation related to acute coronary syndromes (ACS) but remaining controversial .... [here briefly ... mention why the controversy, for example, how does the ACS (signs and symptoms, mentioning associated risk factors, among others), understanding that ACS should be mentioned because at the end of this paragraph emphasis is placed on them, in the same sense, mention in which cases the PCT is used, also taking up everything. As mentioned in more detail in the discussion, this may help to better understand this important clinical approach] ... the aim of the present analysis is to verify in a large population of ACS if procalcitonin remains an independent predictor of infection during an acute cardiac event and if it is related to worse outcome in the follow-up.
Even at the beginning of the Introduction, it should be mentioned in general how common the presence of SCD are, this would give a justification that could round off the importance of studying and reporting these cardiovascular cases in this approach related with procalcitonin.
Reply # 2
We really appreciated the comment. We implement the introduction accordingly. As a matter of fact, patients with MI may often present in the acute phase fever and dyspnea, fatigue, increased white blood cells count. These are all signs and symptoms usually related to heart failure complicating ACS, but if misinterpreted they might induce an overuse of antibiotics in absence of real bacterial infection. In this setting PCT might help the clinician in differential diagnosis and in the identification of bacterial infection. We modified introduction accordingly.
Modified text: section introduction, page 1, lines 44-47
Indeed, patients with MI may often present in the acute phase fever and dyspnea, fatigue, increased white blood cells count. These are all signs and symptoms usually related to heart failure complicating ACS, but if misinterpreted they might induce an overuse of antibiotics in absence of real bacterial infection [1;4].

Reviewer 2 Report
Authors did great job. They were able to show procalcitonin is a useful marker to detect bacterial infection even in ACS patients. This is very useful in day to day practice as both ACS and infection can have SIRS criteria; I would suggest to correct grammatical errors. Few of them are listed below:
-Abstract: There few data about if procalcitonin (PCT) values may predict the risk of bacterial infec- 13
-of inflammation related to acute coronary syndromes (ACS) but remaining controversial 40
-As a matter of fact, only PCT levels increase only in a small per- 43
-non ST-elevation MI, unstable angina) according current guidelines; ii) concentration of 63
-cated electronic case report form (eCRF). Specialized personnel perform this procedure. 81
-same times, this study confirmed that even if PCT tents to be higher in patients who ex- 185
-level were older, presented more frequently acute heart failure, had lower blood pression 210
Author Response
Response to Reviewers
Manuscript: jcm-1524997
Title: Procalcitonin predicts bacterial infection, but not long-term occurrence of adverse events in patients with acute coronary syndrome
Corresponding Author: Rita Pavasini, pvsrti@unife.it
Current revision: #1
Reviewer #2
Comment #1
Authors did great job. They were able to show procalcitonin is a useful marker to detect bacterial infection even in ACS patients. This is very useful in day to day practice as both ACS and infection can have SIRS criteria; I would suggest to correct grammatical errors.
Reply #1
We really thank the Reviewer for the constructive comments. We amended the text following your valuable suggestions.
Comment #2
Few of them are listed below:
-Abstract: There few data about if procalcitonin (PCT) values may predict the risk of bacterial infec- 13
-of inflammation related to acute coronary syndromes (ACS) but remaining controversial 40
-As a matter of fact, only PCT levels increase only in a small per- 43
-non ST-elevation MI, unstable angina) according current guidelines; ii) concentration of 63
-cated electronic case report form (eCRF). Specialized personnel perform this procedure. 81
-same times, this study confirmed that even if PCT tents to be higher in patients who ex- 185
-level were older, presented more frequently acute heart failure, had lower blood pression 210
Reply #2
Thank you for your suggestions. We amened grammatical errors and typos in the text.
Modified text: Section Abstract, page 1, line 15
There are few data about if procalcitonin (PCT) values may predict the risk of bacterial infections and long-term outcome in patients with acute coronary syndromes (ACS)
Modified text: Section Introduction, page 2, line 41
Therefore, PCT has been investigated also in the setting of inflammation related to acute coronary syndromes (ACS) but its role remains controversial.
Modified text: Section Introduction, page 2, line 48 (removed text).
As a matter of fact, PCT levels increase only in a small percentage of uncomplicated ST elevation myocardial infarction (STEMI).
Modified text: Section Methods, page 2, line 68
i) diagnosis of acute coronary syndrome (ST-elevation MI, non ST-elevation MI, unstable angina) according to current guidelines.
Modified text: Section Methods, page 2, line 86
Specialized personnel performed this procedure.
Modified text: Section Discussion, page 7, line 212
At the same time, this study confirmed that even if PCT tents to be higher in patients who experience cardiogenic shock during hospitalization
Modified text: Section Discussion, page 8, line 237
presented more often acute heart failure.
